# Igg Food Antibody Guided Elimination-Rotation Diet Was More Effective than FODMAP Diet and Control Diet in the Treatment of Women with Mixed IBS—Results from an Open Label Study

**DOI:** 10.3390/jcm10194317

**Published:** 2021-09-23

**Authors:** Lucyna Ostrowska, Diana Wasiluk, Camille F. J. Lieners, Mirosława Gałęcka, Anna Bartnicka, Dag Tveiten

**Affiliations:** 1Department of Dietetics and Clinical Nutrition, Medical University of Bialystok, ul. Mieszka I 4B, 15-054 Bialystok, Poland; lucyna.ostrowska@umb.edu.pl; 2Institute of Microecology, ul. Sielska 10, 60-129 Poznan, Poland; clieners@pt.lu (C.F.J.L.); drgalecka@instytut-mikroekologii.pl (M.G.); anna.bartnicka@instytut-mikroekologii.pl (A.B.); 3Lab1 Medical Laboratory, Elias Smiths vei 10, 1337 Sandvika, Norway; dt@lab1.no

**Keywords:** irritable bowel syndrome (IBS), IgG food hypersensitivity, elimination-rotation diet, low-FODMAP diet, classic dietary recommendation, calprotectin

## Abstract

Irritable bowel syndrome (IBS) is a chronic disease with recurrent abdominal pain, disturbed bowel emptying, and changes in stool consistency. We compared the effectiveness of three different dietary treatment plans (G1-FM-low FODMAP diet, G2-IP IgG based elimination-rotation-diet, and as control group, the G3-K control diet recommended by an attending gastroenterologist) in treating patients diagnosed with mixed irritable bowel syndrome. A total of seventy-three female patients diagnosed with a mixed form of irritable bowel syndrome (IBS-M) were enrolled in the study. The diet of each patient in Group 1 (G1-FM) and 2 (G2-IP) was determined individually during a meeting with a dietitian. Patients from Group 3 (G3-K) received nutrition advice from a gastroenterologist. Significant differences in the reduction of IBS symptoms were found between the groups. IBS symptoms as well as comorbid symptoms significantly improved or disappeared completely in the G2-IP group (idiopathic abdominal pain, *p* < 0.001; abdominal pain after a meal, *p* < 0.001; abdominal pain during defecation, *p* = 0.008), while in the G1-FM group, some of the IBS symptoms significantly improved (mucus in stool, *p* = 0.031; bloating, *p* < 0.001). In group G3-K no significant improvement was seen. Based on the results of this open-label study, it was concluded that various dietary interventions in the treatment of IBS-M patients do not uniformly affect the course and outcomes of disease management. Rotation diets based on IgG show significantly better results compared to other diets.

## 1. Introduction

Irritable bowel syndrome (IBS) is a chronic gastrointestinal condition. It is characterized by abdominal pains of different intensity, located usually on the left side of lower abdomen, combined with changes in stool consistency and/or bowel movements [1,2]. Treating patients with IBS is a difficult and complex clinical problem; however, it appears that adequate dietary modifications can comprise an independent element of treatment. Some patients state that nutrition is an important factor influencing the clinical symptoms of IBS [3,4,5]. This view is supported by reports indicating that 20% to 67% of patients with IBS report symptoms occurring after food intake. Patients with IBS often eliminate some food products from their diet. The results of numerous studies indicate that FODMAP carbohydrates (fermentable oligosaccharides, disaccharides, monosaccharides and polyols) in the diet cause or intensify symptoms of IBS and that in contrast, a low FODMAP diet provides relief to patients with IBS [6,7,8,9,10]. It has been shown that a low FODMAP diet was effective in treating the functional symptoms of gastrointestinal conditions in comparison to a diet plan consistent with the recommendations of the UK National Institute for Health and Clinical Excellence (NICE) [9].

In recent years, more and more people believe that the symptoms of IBS may also be due to IgG-dependent food hypersensitivity. In contrast to the FODMAP diet, IgG guided diets implicate the immune system of each individual. Atkinson et al. [11] and Dixon [12] noted that elevated IgG food antibody concentrations in the serum can be a marker of immune activation and a manifestation of delayed food hypersensitivity. They showed that the elimination-rotation diet can be an effective way to alleviate the symptoms of patients with IBS and other symptoms of delayed food hypersensitivities. Other researcher showed that IgG titers to food antigens were higher in patients with IBS than in subjects without IBS [13]. Significant improvements were obtained in a study by avoiding selected IgG positive foods for 6 months [14]. The guidelines from German Society of Digestive and Metabolic Diseases (DGVS) and the German Society of Neurogastroenterology and Motility (DGNM) state that an elimination diet based on high titers of IgG food antibodies can be worth trying in cases of IBS [15]. Other studies showed significant improvement of IBS symptoms together with reduced activity in comorbidities, such as migraines [16,17]. IgG-guided diets may provide individualization of the diet according to individual IgG results, which may increase the effectiveness compared to other diets, where the focus is mainly on the nature of the food, not the response from the patient. Some studies [11,12,13,14,16,17,18,19,20] have proven the beneficial effect of a change to a diet based on IgG antibodies to food not only in cases of IBS but also in other pathologies [21,22,23].

The aim of this report was to compare the effectiveness of an 8-week dietary treatment with three different diet plans (low FODMAP diet, elimination-rotation diet based on IgG-dependent food hypersensitivity test, and a classic diet recommended by an attending gastroenterologist) in treating patients diagnosed with IBS-M.

## 2. Materials and Methods

### 2.1. Participants

Study materials comprised of IBS-M patients diagnosed by an attending physician (gastroenterologist) based on the Rome III criteria with no other digestive tract conditions (e.g., coeliac disease, gastroesophageal reflux disease) that might influence the results of this study. A total of 107 adults applied to be a part of the study. Patients who were using strong opioids or psychotropic drugs and patients who had participated in other clinical studies on digestive tract conditions or who were using a dietary treatment in the 90 days before the study, were excluded from the study. Hence, 73 female patients completed the study. The study was an open-label trial comparing three different dietary treatments of IBS-M. Due to their small number, the group of men was excluded from the study, and 17 women did not agree to be a part of this study, with no reasons given.

The patients included in the study were allocated into three groups. A simple random sampling frame was prepared without returning (in the order of patients reporting). The assignment to the group was as follows: the first patient from the list was selected for Group 1 (G1-FM), every second person was qualified for Group 2 (G2-IP), and every third person was selected for Group 3 (G3-K) (Figure 1). Group 1 (G1-FM) consisted of 26 patients, who, during the 1st appointment, were given a diet plan that was low in fermentable oligosaccharides, disaccharides, monosaccharides, and polyols (FODMAP) for 8 weeks. Each patient received individual dietary advice, materials with an example menu written for 7 days (energy value of the diet—1800–2300 kcal), and a table of products recommended and contraindicated in the FODMAP diet. Group 2 (G2-IP) consisted of 21 patients, who, during the 1st appointment, had their IgG antibody titers tested in response to certain foods in order to find food hypersensitivities. After receiving the results of the test, nutritional counseling regarding the use of an elimination-rotation and exemplary menu (energy value of the diet—1800–2300 kcal) were given to each patient for 8 weeks. IgG-positive foods were eliminated from the diet. All IgG-negative foods were allowed in the rotation diet. There were 26 patients who were classified into Group 3, which was the control group (G3-K), and this group was advised to receive dietary treatment by their attending gastroenterologist for 8 weeks. These patients were given an easily digestible diet consisting of the modification of rational nutrition for healthy people, covering the energy same needs and providing the same amount of nutrients as a normal diet. During periods where the patients were experiencing periods of diarrhea, patients followed an easily digestible diet, limiting fat and the insoluble fraction of dietary fiber (wheat bran, wholemeal bread, thick groats, dark rice, almonds, nuts, poppy seeds, raw vegetables, and fruits with peel and seeds).

In periods where the patients were experiencing constipation, patients used an increased supply of dietary fiber. The diet needed to contain 30–50 g of fiber. Each meal needed to contain at least one fiber-enhanced product, especially soluble fiber (e.g., in the form of fruit, vegetables, puree juices, ground linseed or oatmeal). High-residual products on the menu were introduced gradually due to the fact that some people may not be able to tolerate the a increased supply of fiber during periods of constipation, mainly due to annoying flatulence. In such cases, it was necessary to follow an easily digestible diet with limited raw fiber. Such a diet spares the diseased organ (intestine) and at the same time ensures an adequate amount of residual substances. Increasing the amount of fiber should start with cooked vegetables and puree juices. Then, if those were well tolerated, raw vegetables (finely grated carrot salads, celery, Chinese cabbage, peeled tomatoes, green lettuce) and wholemeal flour products could be introduced.

It was important that patients did not eat too much dietary fiber in one meal nor too rapidly throughout the day. High-residual products were introduced into the diet one by one, carefully observing whether and symptoms appeared after the introduction of certain products. This procedure allowed for the elimination of unfavorable products from the menu without the need to exclude those that were well tolerated.

The diet included an adequate amount of fluids (approx. 2 L per day): non-carbonated mineral waters, fruit teas, herbal teas (mint, chamomile, lemon balm), fruit and vegetable juices.

At the initial and final visits (8 weeks later), each patient was interviewed in detail. The interview questionnaire contained, among others, questions about the most common symptoms of the disease (e.g., abdominal pain, flatulence, bowel movements).

Patients qualified for the study used pharmacotherapy for irritable bowel syndrome, which had been determined earlier by the attending gastroenterologist. Pharmacotherapy with one or more drugs was performed consistently for a minimum of 3 months prior to study enrollment and throughout the study period. A similar drug frequency was found in the three groups: mebeverine was the most frequently used pharmacotherapy (14 people in the G1-FM group, 16 people in the G2-IP group, and 13 people in the G3-K group) followed by trimebutin (14 people in the G1-FM group, 16 people in the G2-IP group, and 16 in the G3-K group) and lubricants and swellings facilitating the passage of stools (12 people in the G1-FM group, 10 people in the G2-IP group, and 12 people in the G3-K group). Simethicone was used less frequently (9 people in the G1-FM group, 6 people in the G2-IP group, 6 people in the G3-K group). There were no significant differences between the studied groups of women in the frequency of using particular drugs (Table 1).

The study was conducted between April 2016 and December 2018. The study was conducted in accordance with the guidelines set out in the 1964 Declaration of Helsinki, and all procedures involving patients were approved by the Bioethics Committee of the Medical University of Bialystok (Poland), approval No. R-I-002/389/2015. The clinical trial was registered at ClinicalTrials.gov, (date of issue: 12 March 2020; date of first registration 25 February 2020), ClinicalTrials.gov ID: NCT04307368.

### 2.2. Methods of Laboratory Testing

#### 2.2.1. Determination of Specific IgG Antibodies Titers against Selected Foods (GROUP 2-G2-IP)

Venous blood serum was collected from the patients. Specific IgG antibodies titers were determined using the ImuPro Complete test (enzyme-linked immunosorbent assay (ELISA) (RIDASCREEN^®^ R-Biopharm, Darmstadt, Germany) according to the manufacturer’s recommendations. A total of 269 foods and IgG antibodies were tested for every patient. Individual tested antibodies and distribution are presented in Appendix A. Obtained values were interpreted based on the concentrations of antibodies expressed in μg/mL: <7.5—not elevated specific IgG concentration, ≥7.5—elevated specific IgG concentration, and ≥20.0—highly elevated specific IgG concentration. Foods that generated elevated and highly elevated IgG concentrations were eliminated from the patient’s diet plan for 8 weeks.

#### 2.2.2. Determination of Fecal Calprotectin Concentration

In order to identify digestive tract inflammation, calprotectin concentration was determined in a stool sample. The test was conducted using the enzyme-linked immunosorbent assay (ELISA) (RIDASCREEN^®^ Calprotectin R-Biopharm) according to the manufacturer’s recommendations. Values ≤ 50 mg/kg of stool were considered as normal values, values > 50 mg/kg of stool are considered elevated values.

### 2.3. Statistical Analyses

IBM^®^ SPSS Statistics version 20.0 (IBM Corp. Released 2011, IBM SPSS Statistics for Windows, Version 20.0 Armonk, NY: IBM Corp.) was used to perform the statistical calculations. Significance of the changes in the frequency of IBS symptoms before and after the prescribed diet plans was tested using the McNemar test. In individual cases, when a certain symptom was not reported by the patient, it was not possible to conduct the test for numerical reasons.

Quantitative variables (i.e., calprotectin concentration) were analyzed using non-parametric tests and were verified using the Shapiro–Wilk test. *p*-values less than 0.05 were considered statistically significant.

## 3. Results

Table 2, Table 3 and Table 4 present the presence of “typical” clinical symptoms, dyspeptic symptoms, and extra-intestinal symptoms in the female IBS-M patients before and after dietary treatment. Patients were asked a closed question with a single choice: “yes” or “no”.

When comparing idiopathic abdominal pain, abdominal pain after a meal, abdominal pain during defecation, and sensation of incomplete defecation before and after the diet plans, statistically significant differences were only found in the case of group G2-IP. (Table 2).

During the 1st examination, mucus in stool was reported by 30.8% of the patients from the group G1-FM, 28.6% of the patients from group G2-IP, and 19.2% of the patients from group G3-K. During the final examination, significant improvement was found in group G1-FM, where only 7.7% (*p* = 0.031) reported mucus in the stool, and in G2-IP, where no patient reported this symptom anymore. However, in group G3-K, the percentage of the patients reporting mucus in the stool increased, but not significantly. (Table 2). In the section concerning typical symptoms connected to the digestive tract, patients were asked whether they experienced constipation, bloating, gurgling sensation, and sensation of gastric fullness. The results proved a significant reduction of these symptoms in group G2-IP. Additionally, after 8 weeks with a low-FODMAP diet, the percentage of patients reporting bloating, gurgling sensation, and sensation of gastric fullness decreased statistically significantly (Table 2). During the final examination, nausea disappeared in groups G1-FM and G2-IP but not in the G3-K group (Table 3).

Extra-intestinal symptoms such as tiredness and weakness, skin lesions, and headaches/migraine were also evaluated before and after the dietary interventions (Table 4). Patients were asked a closed question with a single choice: “yes” or “no”.

## 4. Discussion

IBS is a complex syndrome that probably has several different causes, and a single or generalized treatment will not be efficient. It seems obvious that food plays a major role in the development of IBS, either due to the nature of the food, the immune response from the host, or the microbiota present.

This report includes an analysis of the frequencies of typical IBS symptoms (e.g., idiopathic abdominal pain, abdominal pain after a meal, abdominal pain during defecation, sensation of incomplete defecation, mucus and blood in stool, and bloating), dyspeptic symptoms (nausea, heartburn, and belching), and non-bowel symptoms (constant tiredness, skin lesions, and headaches) occurring before the dietary treatment and after implementing the 8-week elimination-rotation diet plan. After implementing the diet, among patients with IBS-M, a statistically significant reduction of the frequency of the idiopathic abdominal pain was found. Strikingly, a highly significant decrease or complete disappearance of dyspeptic IBS symptoms and comorbidities together with IBS symptoms could only be seen in the G2-IP group, while no differences could be observed in G1-FM and G3-K.

For a long time, it has been known that a low-FODMAP diet is effective in treating the functional symptoms of digestive tract conditions in compared to a diet plan consistent with the recommendations of the UK National Institute for Health and Clinical Excellence (NICE). [7,8,9,24,25,26]. Over 76% of patients reported improvement after introducing the low-FODMAP diet in comparison to the 54% eating according to the NICE recommendations. In these studies, improvements were mainly seen for abdominal pain, abdominal cramps, diarrhea, gas, and bloating. Other symptoms either were not analyzed or did not show any improvement. This seems logical, as the FODMAPs mainly caused an excessive production of gas, leading to discomfort and pain and an increased osmotic effect leading to increased bowel movement and diarrhea. Nevertheless, 30% of the affected patients still suffered from bloating on the FODMAP diet. Similar observations were found in this study, except that both bloating, gurgling sensation, and gastric fullness decreased significantly. Gurgling sensation decreased from 65% to 15%, and gastric fullness decreased from 58% to 11% in the patients on the low FODMAP diet. The patients in the elimination-rotation group had a decrease in bloating from 90% to 9%, in gurgling sensation from 85% to 9%, and in gastric fullness from 90% to 9%. This study shows that a personalized dietary approach is more effective in treating IBS than generalized diet recommendations and that it may even be more effective than a low FODMAP diet. Only the IgG elimination-rotation diet demonstrated significant improvements in all of the monitored IBS symptoms.

IBS is often associated with extra-intestinal comorbid disorders such as migraine [27], asthma, food-, pollen-, and animal allergies; psoriasis; rheumatoid arthritis; and behavior disorders and depression. This provides evidence that different peripheral pathways may be involved in the pathogenesis of certain functional gastrointestinal disorders [28,29]. In recent years, there has been increasing evidence that IBS symptoms are a low-grade inflammatory disease [30,31,32] and may result from or lead to IgG-dependent food hypersensitivities. Our data confirm previous results from different studies showing the effectiveness of an IgG-guided diet in comorbid conditions such as fatigue, headache/migraine, and skin conditions. Identical results were found by Aydinlar [17], who showed that migraine paralleled IBS improvements after a change of diet according to IgG findings. A recent paper [16] confirmed these findings and further showed an increase of serotonin after the elimination diet. This could be an indication of lower inflammation in the gut. Wichers et al. [33] showed that inflammation leads to the depletion of tryptophan via IDO (indole-amine dioxygenase) and thus to lower serotonin levels, promoting fatigue and depression. This confirms the hypothesis that apparently disparate conditions may operate through common pathways, and treatments used exclusively for one of these conditions may prove beneficial for the others [34].

There is still a controversial discussion about the use of IgG tests to detect delayed food sensitivities [35]. The IgG antibodies include four subclasses IgG_1-4_. They efficiently opsonize pathogens, enabling their absorption by phagocytes and activating the complement system, except IgG4 [36,37]. IgG_4_ are antibodies released in response to IL-10 (anti-inflammatory cytokine), an antibody connected to the Th2 immune response in the process of desensitization in type-I allergies (IgE-independent). Since the discovery of the Th17 linage [38], IgG must be looked at from a different point of view than before. IgG is not only the immunoglobulin that protects us against foreign infectious agents but is now also recognized as a mediator of inflammation and is responsible for auto-immune diseases. The balance of Th17 and Treg is largely responsible for the pro-inflammatory conditions in our body. Both are expressed in peripheral tissue and in the gut in particular [39]. Alterations of the microbiome [40] and leaky gut induce the activation of the Th17 mediated immune response and the production of pro-inflammatory IgG antibodies against food and other potential harmful antigens present in the gut [40]. The publication by Jönsson et al. [18] demonstrated the pathological pathways of IgG inducing an inflammatory reaction via the platelet activation factor (PAF) with the participation of IgG receptors and neutrophils. They also showed and reproduced previous scientific findings [41,42,43] that IgG can be implicated in anaphylaxis.

It is difficult to assess low-grade inflammation in the gut. Among the available diagnostic markers, calprotectin is currently one of the best-known markers indicating mucosa inflammation and changes in the inflammation intensity [44,45,46,47,48,49,50]. In this paper, it was assumed that serious intestinal inflammation was diagnosed at the fecal calprotectin concentration of >50 mg/kg of stool. During the first examination, no statistically significant differences were found in calprotectin concentrations between the compared groups of patients, and the values were low. These findings suggest that the included patients suffered from low-grade inflammation and were suitable for diet alteration as the best choice of treatment.

The main limitations of this study are the open-labeled nature and the low number of participants in the study. Another weakness is that the patients consist of females only. Further, all of the patients were not tested for IgG food antibodies: only those in the G2 group were. It would be helpful to compare what the common IgG food hypersensitivities between the groups were. It would also be of interest to know which foods the patients were ingesting before they entered the study. These limitations could impact or influence the interpretation of the findings from our research.

## 5. Conclusions

This study shows that a personalized dietary approach is more effective in treating IBS-M than generalized diet recommendations. Only the IgG elimination-rotation diet could demonstrate significant improvements in all of the monitored IBS-M symptoms as well as extra-intestinal symptoms. None of the diets have shown 100% effectiveness. By applying an IgG-guided elimination diet, some FODMAPs are automatically removed as well, depending on which foods have to be avoided. One possible strategy could be to start with the elimination-rotation diet, as it was proven to be the more effective diet in this open study, and in cases of persistent symptoms, it could be combined with a low-FODMAP diet. Claims that IgG food antibodies only reveal exposure to food and not intolerance should be reinvestigated in larger double-blinded studies.

## Figures and Tables

**Figure 1 jcm-10-04317-f001:**
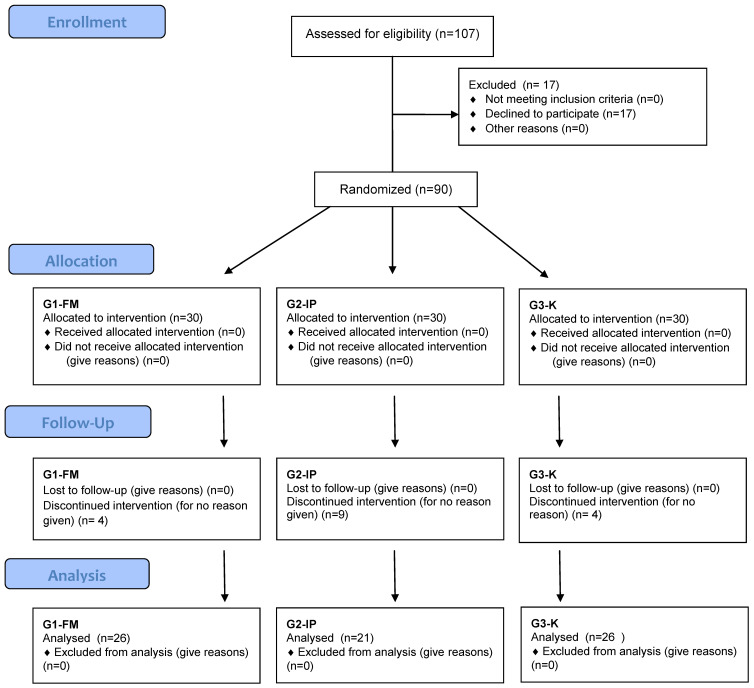
Randomization schedule.

**Table 1 jcm-10-04317-t001:** Demographic characterization of the patients.

Selected Demographic Features	G1-FM (*n*= 26)	G2-IP (*n* = 21)	G3-K (*n* = 26)	*p*
**AGE**				
Mean age	42.70 ± 16.70	40.60 ± 14.50	41.70 ± 13.40	0.931 *
Median	46.00	46.00	43.50	
Total mean age		41.70 ± 14.80		
**BODY WEIGHT (kg)**				
Mean 1st examination	66.73 ± 13.35	63.14 ± 9.13	73.77 ± 19.85	0.126 *
Mean 2nd examination	66.73 ± 13.35	63.14 ± 9.13	73.77 ± 19.85	0.090 *
*p*	0.531	0.055	0.680	
**BMI (kg/m^2^)**				
Mean 1st examination	24.12 ± 4.58	23.81 ± 3.72	26.50 ± 6.93	0.341 *
Mean 2nd examination	24.09 ± 4.43	23.54 ± 3.59	26.40 ± 6.86	0.324 *
*p*	0.918 ***	**0.038 *****	0.684 ***	
**SMOKING TOBACCO**				
Smokes	0	4 (19.00%)	4 (15.40%)	0.071 *
Does not smoke	19 (73.10%)	10 (47.60%)	19 (73.10%)	
Expeller	7 (26.90%)	7 (33.30%)	3 (11.50%)	
**LEVEL OF EDUCATION**				
Vocational	2 (7.70%)	2 (9.50%)	2 (7.70%)	0.078 **
Secondary	7 (26.90%)	11 (52.40%)	4 (15.40%)	
Higher	17 (65.40%)	8 (38.10%)	20 (76.90%)	
**PLACE OF RESIDENCE**				
Village	3 (11.50%)	3 (14.30%)	1 (3.80%)	0.155 **
Town	0	3 (14.30%)	1 (3.80%)	
City	23 (88.50%)	15 (71.40%)	24 (92.30%)	
**IBS DURATION**				
<5 years	11 (42.30%)	12 (57.10%)	11 (42.30%)	0.849 **
6–10 years	9 (34.60%)	5 (23.80%)	9 (34.60%)	
>11 years	6 (23.10%)	4 (19.00%)	6 (23.10%)	

*p* * Kruskal Wallis test. *p* ** Pearson’s chi-square independence test. Dependence test. *p* *** Wilcoxon’s order of pairs test (comparison of studies 1 vs. 2). G1-FM—the FODMAP group. G2-IP—the elimination-rotation group. G3-K—the control group.

**Table 2 jcm-10-04317-t002:** Frequency of IBS symptoms in studied patients before and after dietary treatment.

Symptoms		G1-FM			G2-IP			G3-K		
		1st Examination	2nd Examination	*p*	1st Examination	2nd Examination	*p*	1st Examination	2nd Examination	*p*
Idiopathic abdominal pain	N	15	11	0.125	16	2	<0.001	16	14	0.500
	%	57.7	42.3		76.2	9.5		61.5	53.8	
Abdominal pain after a meal	N	11	6	0.063	14	2	<0.001	14	12	0.625
	%	42.3	23.1		66.7	9.5		53.8	46.2	
Abdominal pain during defecation	N	5	2	0.250	9	1	0.008	6	6	1.000
	%	19.2	7.7		42.9	4.8		23.1	23.1	
Sensation of incomplete defecation	N	13	10	0.250	13	2	0.001	14	15	1.000
	%	50.0	38.5		61.9	9.5		53.8	57.7	
Mucus in stool	N	8	2	0.031	6	0	*	5	6	1.000
	%	30.8	7.7		28.6	0.0		19.2	23.1	
Blood in stool	N	3	0	*	2	0	*	2	2	1.000
	%	11.5	0.0		9.5	0.0		7.7	7.7	
Difficulty to defecate (constipations)	N	11	7	0.219	14	4	0.002	19	17	0.500
	%	42.3	26.9		66.7	19.0		73.1	65.4	
Bloating	N	22	7	<0.001	19	2	<0.001	24	22	0.500
	%	84.6	26.9		90.5	9.5		92.3	84.6	
Gurgling sensation	N	17	4	<0.001	18	2	<0.001	21	19	0.500
	%	65.4	15.4		85.7	9.5		80.8	73.1	
Gastric fullness	N	15	3	<0.001	19	2	<0.001	22	19	0.250
	%	57.7	11.5		90.5	9.5		84.6	73.1	

McNemar *p* test. * Conducting the test was impossible because the symptom was not reported by any patient during the 2nd examination. G1-FM—the FODMAP group. G2-IP—the elimination-rotation group. G3-K—the control group. N—number of patients.

**Table 3 jcm-10-04317-t003:** Frequency of dyspeptic IBS symptoms in studied patients before and after dietary treatment.

Symptoms	G1-FM				G2-IP			G3-K		
	1st Examination		2nd Examination	*p*	1st Examination	2nd Examination	*p*	1st Examination	2nd Examination	*p*
Nausea	N	6	0	*	7	0	*	9	9	1.000
	%	23.1	0.0		33.3	0.0		34.6	34.6	
Heartburn	N	2	2	1.000	7	1	0.031	5	4	1.000
	%	7.7	7.7		33.3	4.8		19.2	15.4	
Belching	N	5	4	1.000	6	0.0	*	7	7	1.000
	%	19.2	15.4		28.6	0.0		26.9	26.9	

McNemar *p* test. * Conducting the test was impossible because the symptom was not reported by any patient during the 2nd examination. G1-FM—the FODMAP group. G2-IP—the elimination-rotation group. G3-K—the control group.

**Table 4 jcm-10-04317-t004:** Frequency of extra-intestinal symptoms in patients before and after dietary treatment.

Symptoms		G1-FM			G2-IP			G3-K		
		1st Examination	2nd Examination	*p*	1st Examination	2nd Examination	*p*	1st Examination	2nd Examination	*p*
Constant tiredness and weakness	N	5	4	1.000	7	1	0.031	9	8	1.000
	%	19.2	15.4		33.3	4.8		34.6	30.8	
Skin conditions	N	0	0	*	4	0	*	1	1	1.000
	%	0.0	0.0		19.0	0.0		3.8	3.8	
Headaches/migraines	N	3	3	1.000	3	0	*	5	4	1.000
	%	11.5	11.5		14.3	0.0		19.2	15.4	

McNemar *p* test. * Conducting the test was impossible because the symptom was not reported by any patient during the 2nd examination. G1-FM—FODMAP group. G2-IP—the elimination-rotation group. G3-K—the control group.

## Data Availability

The datasets are available from the corresponding author on reasonable request.

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
