# Peer review of "Igg Food Antibody Guided Elimination-Rotation Diet Was More Effective than FODMAP Diet and Control Diet in the Treatment of Women with Mixed IBS—Results from an Open Label Study"

_jcm, 2021, doi:10.3390/jcm10194317_

Round 1

Reviewer 1 Report

In this study, Ostrowska et al. compared the effectiveness of three different diet plans of 8-weeks in treating patients with mixed irritable bowel syndrome. The study extends the evidence of the utility of the elimination-rotation diet based on the IgG-dependent food hypersensitivity test. In my opinion, the strength of the study includes the number of antibodies tested. The study's limitations include the patients, which all are females, the number of patients, the type of IBS (only mixed form), and the time of the diet plans.

I have some recommendations that can improve the manuscript:   

First page Line 22. The citation includes the journal nutrients, which is not correct; please check.

What was the reason for including only IBS-M?

How was celiac disease excluded? The majority of the 21 patients included in group 2 responded to gluten.    

The data generated by testing IgG is interesting; however, the authors did not perform the same tests in the three groups. Also, having the antibody data after the diet plans in the three groups would increase the interpretation, probably the antibodies to the foods removed from the diet decreased.    

Author Response

The overall goal of this study was to establish the most effective dietary treatment regimen for patients with mixed IBS. Patients diagnosed with IBS-M were selected for examination by a Gastroenterologist in accordance with the current standards of disease diagnosis. According to the latest guidelines, diagnosis of irritable bowel syndrome should be based on clinical symptoms. In diarrhea IBS, celiac disease should always be ruled out when it is necessary to differentiate the disease from inflammatory bowel disease. I am aware that the study has some limitations such as the small number of participants and the fact that only G2 patients were tested for the presence of IgG antibodies in food. Therefore, these limitations are presented in the last paragraph of the discussion. The study was a pilot study, due to the costs associated with the test for the presence of IgG antibodies (ImuPro), only patients from the G2 group had this study (the cost of one test is 410 Euro). When planning the next study, the above elements will be taken into account.

Reviewer 2 Report

Reviewer's report

Title: Igg food antibody guided elimination-rotation diet was more effective than FODMAP diet and control diet in the treatment of women with mixed IBS - results from an open label study

Version: 1st version

Date: 16.07.2021

Reviewer's report:

Ostrowska et al. presented a well-designed study of good scientific quality. The study compares the effectiveness of dietary treatment of 3 different diet plans (low FODMAP, IgG based elimination and standard diet recommended by gastroenterologists as control) in treating patients with mixed form of irritable bowel syndrome. Their report includes the analysis of the frequencies of typical irritable bowel syndrome symptoms, dyspeptic symptoms and non-bowel symptoms occuring before and after 8 week dietary treatment. It is interesting to see that only IgG elimination-rotation diet could demonstrate significant improvements in all monitored symptomes. However as authors also discussed and mentioned as limitations of the study, due to open label nature, low number of participants, female gender only and the fact that not all the participants were tested for IgG food antibodies; the results should be interpreteted causiously. There is also a fact that as authors mentioned in conclusion with IgG rotation diet the low FODMAP foods were also possibly eliminated. 

In the end, the study is well-designed, presented clearly, the results are scientifically presented, the manuscript is easy to follow.

The only recommendation I would do is maybe to add Table 1 as a supplementary since it is way too long and makes reader to lose the focus when reading the publication. However it is up to authors to decide, the second smal thing that caught my eye is on table to column 2 G1-FM, n=26 title missing the braclet ( ) that other columns have.

 Author Response

Thank you for the positive review of the article and the suggestion for improvement in table 2 "(n = 26)".

Reviewer 3 Report

This is an interesting study aiming to compare head-to-head different approaches to the treatment of IBS. Seventy-three female patients were allocated to three groups, one with FODMAP restriction, one with dietary restriction based on IgG against a large range of different foods, and the third with advise from a gastroenterologist. Although of interest, there are some weak aspects:

  1. The selection of patients is not very clear. Are these referred from general practice?
  2. The randomisation procedure is unclear.
  3. The different groups were given advise but the open nature of this intervention is an obvious source of problems, as the expectation of the study members may heavily influence the outcomes.
  4. Table 1 is an overview of all the test done. In total, elevated and higly elevated values were seen for 170 items in serum from the 21 individuals in this group. Thus it is likely that the intervention was very complex, but unfortunately, no details are given. The structure of the info could be made more clear, not he least can many of the negative values be omitted. With som many tests, one further wonders about the reproducability on same serum and the persistence of positive values in each patient.
  5. No details on the FODMAP intervention and the advise by the gastroenterologist are given. The reader may be interested in details.                

Author Response

Yes, patient selection relates to general practice.

The randomization of the study consisted in the fact that the composition of the groups was selected at random (random distribution of the respondents to the comparative groups).

Agree that various groups have been advised that the open-minded nature of this intervention is an obvious source of problems as the expectations of study members can have a large impact on the results.

With so many tests, one may further wonder about the reproducibility on the same serum and the persistence of positive values in each patient. This will be taken into account when planning further research.

A diet low in fermentable oligosaccharides, disaccharides, monosaccharides and polyols (FODMAPs) is used to relieve symptoms of irritable bowel syndrome (IBS). Patients were given standard guidelines regarding this diet and a written menu. On the other hand, the gastroenterologist provided dietary advice in accordance with the current recommendations regarding the diet in IBS-M.

Round 2

Reviewer 1 Report

I did not see an improvement in the manuscript after the comments from the reviewers; however, the data is found attractive, and nothing else could be changed given that the study's design was not the most appropriate.

Author Response

Valuable comments from reviewers have been written in the manuscript. 

This manuscript is a resubmission of an earlier submission. The following is a list of the peer review reports and author responses from that submission.

Round 1

Reviewer 1 Report

In this study, Ostrowska et al. determined the effectiveness of three different diet plans of 8-weeks in treating patients with mixed irritable bowel syndrome. The study extends the evidence of the utility of the elimination-rotation diet based on IgG-dependent food hypersensitivity test as a treatment in patients with irritable bowel syndrome.

I have some recommendations that can improve the manuscript:   

Is there a specific reason to include only IBS-M?

How was celiac disease excluded? The majority of the 21 patients included in group 2 responded to gluten.    

The data generated by testing IgG is interesting; however, the authors did not perform the same tests in the other two groups (I appreciate that the authors included it as a limitation). Also, having the antibody data after the diet plans in the three groups would increase the interpretation, probably the antibodies to the foods removed from the diet decreased.    

What was the reason for the 8-week treatments? Why not keep the treatment even longer if it can relieve the symptoms? 

Which program was used to perform the statistical test?

Author Response

Is there a specific reason to include only IBS-M?

The overall goal of this study was to establish the most effective dietary treatment regimen for patients with mixed IBS. Conducting the study in other subtypes of irritable bowel syndrome (diarrhea, constipation form) is a limitation of the study.

How was celiac disease excluded? The majority of the 21 patients included in group 2 responded to gluten.    

The authors considered this as a work limitation. According to the latest guidelines, diagnosis of irritable bowel syndrome should be based on clinical symptoms. In diarrhea IBS, celiac disease should always be ruled out when it is necessary to differentiate the disease from inflammatory bowel disease.

The data generated by testing IgG is interesting; however, the authors did not perform the same tests in the other two groups (I appreciate that the authors included it as a limitation).

The authors considered this as a work limitation.

Also, having the antibody data after the diet plans in the three groups would increase the interpretation, probably the antibodies to the foods removed from the diet decreased.   

Since IgG has a very long half-life, I believe it would be difficult to detect any major decrease only after 8 weeks. IgA antibodies should be included, then a much faster reduction of antibodies can be noticed after the diet.

What was the reason for the 8-week treatments? Why not keep the treatment even longer if it can relieve the symptoms? 

The basic method of treating food hypersensitivity is the use of an elimination-rotation diet aimed at temporarily or permanently excluding sensitizing foods from the menu. At the same time, in place of the eliminated products, products with a similar nutritional value should be introduced, so that the diet is balanced and provides all the necessary nutrients, in the amount appropriate to the demand. Dietary guidelines for the use of an elimination-rotation diet are based on two important components: elimination and rotation (literature attached). Strict elimination is based on the elimination of all food products with elevated and significantly elevated IgG antibody concentration. Strict avoidance of food products with elevated and significantly elevated levels of IgG antibodies allows to limit or even stop inflammatory processes. Initial elimination takes 5 to 8 weeks. Then rotation is about eating foods that are allowed to be eaten. If a patient consumes a certain product on one day, he must avoid consuming them for the next 3 days. This will help the body eliminate current Type III (IgG-mediated) food allergies while reducing the chances of causing new allergies. Therefore, I believe 8 weeks is good for elimination diets. In the assumptions of our study, we wanted to check the effectiveness of the introduced dietary treatment for 8 weeks.

Scarlata K. Low-Fodmap 28-Day Plan: A Healthy Cookbook with Gut-Friendly Recipes for IBS Relief.  Rockridge Press, Berkeley, California 2014.

Canonica GW, Pawankar R, Holgate ST i wsp. WAO: White Book on Allergy, World Allergy Organization 2011.

Frank M, Ignyś I, Gałęcka M i wsp. Alergia pokarmowa IgG-zależna i jej znaczenie w wybranych jednostkach chorobowych. Ped Pol. 2013, 88(4): 252-257.

Which program was used to perform the statistical test?

IBM® PSS Statistics version 20 was used to perform statistical calculations. The statistical hypotheses were verified assuming the value of the 1st type statistical error α= 0.05.

Reviewer 2 Report

The study is interesting to the readers in my view and encouraging future research around the role of IgG based diets for food intolerances. I have just a few comments as follows:

  1. In line 25, it is not clear what is meant by “unlikely effective”. Do the authors mean “unequally effective”?
  2. Providing information about the control diet, which was described as a classic too, would be helpful for the reader to compare.
  3. I couldn’t observe any citations around the IgG response level categorization as non-elevated, elevated, highly elevated. What is the evidence for this categorisation otherwise?
  4. Line 100 seems to contradict line 104-105 as the registration should preceded the conduction of the study.
  5. My understanding is that the study was based on IBS-M subtype, but the conclusion generalizes to all IBS; I think the language of the conclusion needs to be moderated to reflect the fact that the sample was not representative of all IBS types. The abstract language seems sensible to me and agrees with the actual findings unlike the conclusions.
  6. Please revise line 238, “The publication Jönnson et al [18] demonstrated…”.

Author Response

  1. In line 25, it is not clear what is meant by “unlikely effective”. Do the authors mean “unequally effective”?

Yes, the authors mean that:

Based on the results of this open-label study, various dietary interventions in the treatment of IBS-M patients were found to be unequally effective.

Has been changed in the text.

  1. Providing information about the control diet, which was described as a classic too, would be helpful for the reader to compare.

Agree, we will consider these suggestions in future research.

  1. I couldn’t observe any citations around the IgG response level categorization as non-elevated, elevated, highly elevated. What is the evidence for this categorisation otherwise?

The titre of specific IgG1-3 antibodies was tested using the ImuPro Complete test (RIDASCREEN® R-Biopharm), in which 269 nutrients and IgG1-3 antibodies are tested. The ImuPro Complete test is for in vitro diagnostic use; It is based on the ELISA (enzyme immunoassay) method and allows to detect specific IgG1-3 antibodies against particular food antigens in human serum.

Based on the concentration of antibodies expressed in μg / ml, the obtained values were interpreted:

Response level<7.5 μg/ml IgG - not elevated

Response level ≥7.5 μg/ml IgG - elevated

Response level ≥20.0 μg/ml IgG - highly elevated

Table 1 shows the presence of specific IgG1-4 antibodies in response to 269 foods in women in the G2-IP group (n-21).

  1. Line 100 seems to contradict line 104-105 as the registration should preceded the conduction of the study.

There is no obligation to register a clinical trial in Poland. The research protocol was approved by the Bioethics Committee of the Medical University of Bialystok. In order to meet the possible requirements of journals, we decided to register it in the ClinicalTrials.

  1. My understanding is that the study was based on IBS-M subtype, but the conclusion generalizes to all IBS; I think the language of the conclusion needs to be moderated to reflect the fact that the sample was not representative of all IBS types. The abstract language seems sensible to me and agrees with the actual findings unlike the conclusions.

This was changed to IBS-M (lines 261, 262).

  1. Please revise line 238, “The publication Jönnson et al [18] demonstrated…”.

Line 238 is now line 240 (text has been added to the work) I corrected the name of Jönsson in the sentence „The publication Jönsson et al [18] demonstrated..."

Reviewer 3 Report

The authors in their work demonstrate the efficacy of the IgG food antibody guided elimination-rotation diet in the improvement of IBS symptoms.

The study presents some limitations such as the low number of participants and mainly the fact that only the patients of the G2 group were tested for IgG food antibodies.

Overall the paper is well written and very interesting.

Author Response

The study presents some limitations such as the low number of participants and mainly the fact that only the patients of the G2 group were tested for IgG food antibodies.

I am aware that the study presents some limitations such as the small number of participants and the fact that only G2 patients were tested for the presence of IgG antibodies in food. Therefore, these limitations are presented in the last paragraph of the discussion. When planning the next study, the above elements will be taken into account.
